# β2-Adrenergic Signalling Promotes Cell Migration by Upregulating Expression of the Metastasis-Associated Molecule LYPD3

**DOI:** 10.3390/biology9020039

**Published:** 2020-02-22

**Authors:** Michael Gruet, Daniel Cotton, Clare Coveney, David J. Boocock, Sarah Wagner, Lucie Komorowski, Robert C. Rees, A. Graham Pockley, A. Christopher Garner, John D. Wallis, Amanda K. Miles, Desmond G. Powe

**Affiliations:** 1John van Geest Cancer Research Centre, Nottingham Trent University, Nottingham NG11 8NS, UK; michaelgruet@hotmail.co.uk (M.G.); Daniel_Cotton@hotmail.co.uk (D.C.); Clare.Coveney@ntu.ac.uk (C.C.); david.boocock@ntu.ac.uk (D.J.B.); sarah.wagner@ntu.ac.uk (S.W.); lucie.komorowski@gmail.com (L.K.); robert.rees@ntu.ac.uk (R.C.R.); graham.pockley@ntu.ac.uk (A.G.P.); des.powe@talktalk.net (D.G.P.); 2School of Science and Technology, Nottingham Trent University, Nottingham NG11 8NS, UK; christopher.garner@ntu.ac.uk (A.C.G.); john.wallis@ntu.ac.uk (J.D.W.); 3Department of Cellular Pathology, Queen’s Medical Centre, Nottingham University Hospitals NHS Trust, Derby Road, Nottingham NG7 2UH, UK

**Keywords:** breast cancer, β2-adrenoceptor, beta-blockers, LYPD3

## Abstract

Metastasis is associated with poor prognosis in breast cancer. Although some studies suggest beta-blockers increase survival by delaying metastasis, others have been discordant. This study provides both insights into the anomalous findings and identifies potential biomarkers that may be treatment targets. Cell line models of basal-type and oestrogen receptor-positive breast cancer were profiled for basal levels of adrenoceptor gene/protein expression, and β2-adrenoceptor mediated cell behaviour including migration, invasion, adhesion, and survival in response to adrenoceptor agonist/antagonist treatment. Protein profiling and histology identified biomarkers and drug targets. Baseline levels of adrenoceptor gene expression are higher in basal-type rather than oestrogen receptor-positive cancer cells. Norepinephrine (NE) treatment increased invasive capacity in all cell lines but did not increase proliferation/survival. Protein profiling revealed the upregulation of the pro-metastatic gene Ly6/PLAUR Domain-Containing Protein 3 (LYPD3) in norepinephrine-treated MDA-MB-468 cells. Histology confirmed selective LYPD3 expression in primary and metastatic breast tumour samples. These findings demonstrate that basal-type cancer cells show a more aggressive adrenoceptor-β2-activated phenotype in the resting and stimulated state, which is attenuated by adrenoceptor-β2 inhibition. This study also highlights the first association between ADRβ2 signalling and LYPD3; its knockdown significantly reduced the basal and norepinephrine-induced activity of MCF-7 cells in vitro. The regulation of ADRβ2 signalling by LYPD3 and its metastasis promoting activities, reveal LYPD3 as a promising therapeutic target in the treatment of breast and other cancers.

## 1. Introduction

Metastasis is critical in the progression of breast cancer and is frequently associated with poor prognosis. Although strategies that inhibit metastasis will increase progression-free survival (PFS), the identification of therapeutic druggable targets that prevent metastasis remain in its infancy [1]. Approaches for preventing metastasis require drugs that have *cytostatic* rather than *cytotoxic* properties, and are principally aimed at suppressing progression along the multistep metastasis pathway [2]. 

The repurposing of beta-adrenergic receptor antagonists (beta-blockers) as an adjuvant therapy for the treatment of breast cancer has been proposed on the basis of their anti-metastatic properties [3,4,5]. In vitro and in vivo models have demonstrated propranolol-induced inhibition of cancer cell signalling pathways decreases cell adhesion, migration, invasion, extravasation and colonisation in distant tissues including bone [6,7,8], thereby leading to reduced metastasis [9]. These pathways are triggered by catecholamine hormones such as norepinephrine acting on beta (β-) adrenergic G-protein coupled receptors (GPCR) expressed on breast cancer cells. Epidemiology studies observing the therapeutic potential of beta-blockers for treating breast cancer have revealed an association between (coincidental) beta-blocker usage and survival benefits [10,11,12]. The clinical evaluation of propranolol as a neoadjuvant or perioperative treatment for breast cancer is on-going [13,14,15,16]. However, a recent contradictory study has reported no benefit between prescribed beta-blockers and survival [17], whereas a different study using the basal-type MDA-MB-231 breast cancer cell line model showed that beta-adrenergic receptor (ADRβ2) agonism (rather than antagonism) inhibited tumour proliferation [18]. Further studies are required to explain these discordant findings, which could result from variance in (a) in vitro cell line models; (b) patient cohort selected in pre-clinical studies; (c) pharmacologic selectivity of prescribed beta-blockers. 

In this study, adrenoceptor expression and β2-adrenoceptor-mediated metastasis-associated cell behaviour were examined in three frequently used in vitro cell line models of ‘stress-induced’ triple-negative basal-type breast cancer and compared to a popular oestrogen-positive cell line model. β2-adrenoceptor-induced proteomic changes were assessed to better understand ADR-mediated cancer pathways, and provide biomarker and therapeutic treatment target identification. The study reveals complex and distinct differences between the cell lines and also identified a link between ADRβ2 signalling and LYPD3; revealing LYPD3 as a potential key mediator in ADRβ2 driven metastasis. 

## 2. Results

### 2.1. Basal-Type Breast Cancer Cell Lines Express Higher Levels of Functional β2-Adrenoceptor and Their Survival Is Not Significantly Altered Following Non-Selective ADRβ Activation

The steady state mRNA expression of each ADR subtype was assessed in unstimulated breast cancer cell lines. β2-adrenoceptor gene expression was highest in the unstimulated MDA-MB-231 basal cell line, followed by MDA-MB-468 and BT-549. Negligible expression was observed in the ER-positive MCF-7 cell line (Figure 1A). To evaluate the cell surface expression of the selected ADRs, flow cytometry was performed. The level of membranous ADRβ2 expression was highest in the unstimulated basal cell line MDA-MB-468 although levels were very similar between this cell line and MDA-MB-231 and MCF-7. BT-549 cells expressed the lowest levels of ADRβ2 (Figure 1B). These results show that the cell lines, in our hands, express ADRs at both the mRNA and protein levels. All cell lines treated with the non-selective ADRβ agonist isoproterenol showed elevated accumulation of intracellular cAMP (MDA-MB-231 > MDA-MB-468 > BT-549 > MCF-7) (Figure 1C), confirming functional ADR. Furthermore, simultaneous treatment with norepinephrine and the ADRβ2 selective antagonist ICI-118,551 had no significant effect on cell survival at therapeutically relevant concentrations compared to treatment of the cells with norepinephrine alone (Appendix A). These results demonstrate that any change in cell migration/invasion observed following ADR agonism/antagonism is not due to the compounds affecting cell survival. 

### 2.2. Substrate-Modulated Cell Adhesion/Migration/Invasion Responses Are ADRβ-Dependent

The ability of ADR agonist and antagonist compounds to modulate different parameters associated with metastasis were assessed. 

#### 2.2.1. Cell Adhesion

Cell-cell and cell-substrate adhesion characteristics play a pivotal role in many of the key steps of the metastatic cascade and in the metastatic cascade, cells alter their adhesive capacity during transmigration, intravasation and extravasation. The observed increases in adhesive capacity induced by norepinephrine and isoproterenol are mediated by the β1-integrins. Strell et al. also concluded that two possible routes of treatment for the inhibition of MDA-MB-231 cell adhesion to lung epithelial cells is to either block β2-adrenoceptor signalling or to block the action of the β1-integrins [7]. 

Although the use of tissue culture plastic can provide an insight into the adhesive capacity of cells, its use does not appropriately reflect the extracellular matrix proteins that make up the surfaces of the tumour microenvironment and other important areas of the body that cells may encounter during the metastatic cascade. Fibronectin, vitronectin and collagen I are among the most abundant extracellular matrix proteins and are among the key ligands of the crucial adhesion-regulating integrins, therefore, the influence norepinephrine has on breast cancer cells adhering to these substrates was investigated. 

Simultaneous treatment with norepinephrine and the ADRβ selective antagonist ICI-118,551 reduced the adhesion in all four cell lines grown on collagen I and vitronectin coated surfaces (*p* < 0.05). The adhesion of the basal-type MDA-MB-468 and MDA-MB-231 cell lines to fibronectin-coated surfaces was also reduced (*p* < 0.01) (Figure 2A and Appendix A). These results indicate that the adhesion of breast cancer cells can vary according to the surface on which they are grown. 

#### 2.2.2. Cell Migration

The ADRβ selective antagonist ICI-118,551 completely abrogated the enhanced migration of MDA-MB-468, BT-549 and MCF-7 cells induced by norepinephrine treatment, as assessed using scratch migration assays (Appendix A). When the migration of cells through an 8 µm porous membrane was assessed, norepinephrine caused a significant increase in the migration of MDA-MB-468 cells (*p* < 0.01) and this increase was completely reversed by the concomitant administration of ICI-118,551 (*p* < 0.01) (Figure 2B). Interestingly, norepinephrine treatment significantly reduced the migration of MDA-MB-231 cells when grown on both uncoated plastic (Appendix A) and when migrating through a porous membrane (*p* < 0.01) (Figure 2B). Treatment with ICI-118,551 alone reduced migration, albeit non-significantly (Figure 2B). Whilst there was an increase in migration of BT-549 and MCF-7 cells following norepinephrine stimulation and a subsequent decrease following concomitant ICI-118,551 administration, this was not statistically significant. These observations were confirmed and were statistically significant in our scratch assay data (Appendix A), which leads us to believe that this technique for measuring cell migration may not be the best method for every cell line. These findings reinforce the need for investigating the invasive behaviour of cells using a more biologically relevant model and whilst in vivo models would answer these questions, it is important to try and identify a relevant in vitro model before animal models are used.

#### 2.2.3. Cell Invasion

Norepinephrine stimulation alone significantly increased the invasive capacity of all breast cancer cell lines through a basement protein-coated membrane (*p* < 0.05). Furthermore, treatment with the ICI-118,551 antagonist significantly decreased the invasive capacity of MDA-MB-468, MDA-MB-231, BT-549 and MCF-7 cell lines (*p* < 0.05) (Figure 2C), suggesting that our cell line models are able to recapitulate what others in the literature have observed and provide a sound model to investigate proteomic changes following ADR agonism/antagonism [19]. 

### 2.3. Protein Expression Changes in MDA-MB-468 and MDA-MB-231 Cells

To further investigate the potential role of the ADRβ2 in breast cancer cell migration, we performed mass spectrometry analyses of protein extracts taken from the lysate and media (secretome) of MDA-MB-468 and MDA-MB-231 cells treated with norepinephrine, isoproterenol and ICI-118,551 (Appendix A). These cell lines were chosen for the following reasons: increased levels of ADRβ2 mRNA expression, the cells displayed opposing effects following norepinephrine stimulation and the cell lines were derived from the triple negative phenotype, which is of importance for developing new potential treatment regimens or repurposing current therapeutics. Differentially expressed proteins for MDA-MB-468 and MDA-MB-231 cells secretome and lysate are shown in Table 1 and Table 2. One thousand one hundred and sixty eight (MDA-MB-468 secretome), 2715 (MDA-MB-468 lysate), 2635 (MDA-MB-231 secretome) and 3117 (MDA-MB-231 lysate) proteins were quantified by SWATH-MS (Sequential Window Acquisition of All Theoretical Mass Spectra – Mass Spectrometry) and processed using OneOmics (Sciex, Framingham, MA, USA). The analysis revealed several differentially expressed proteins within each cell line; however, the similarity between the cell lines in terms of shared proteins was minimal with the following shared proteins only observed in the norepinephrine and ICI-118,551 treated cells; H4 (P02805) and TSP (P07996). This is not unsurprising given the differences in magnitude of response following stimulation of these cells with ADR agonists/antagonists. 

Of significant interest was one protein, LYPD3, whose increased levels were found in the lysate (2.235 fold change) and secretome (1.577 fold change) of MDA-MB-468 cells following norepinephrine treatment (Table 1, Appendix A) and in the lysate of MDA-MB-468 isoproterenol treated cells (Table 1 and Appendix A). Concomitant treatment with norepinephrine and ICI-118,551 produced a smaller increase in LYPD3 levels in the secretome of MDA-MB-468 (1.308 fold change) cells compared to norepinephrine stimulation alone (Table 1).

### 2.4. Increased LYPD3 Protein Is Exclusively Expressed in Primary and Metastatic Breast Cancer

The association between LYPD3 protein expression with tumour malignancy, patient age, tumour grade and stage, TNM, oestrogen receptor and HER2 status are shown in Table 3. LYPD3 protein staining was localised in the cytoplasm of breast adenocarcinoma cells, with occasional intense staining in cellular membranes (Figure 3A). LYPD3 protein expression was exclusively localised in primary (13.1% positive) and metastatic (17% positive) breast adenocarcinoma tissues, with no difference between the two groups (*p* = 0.171). LYPD3 protein was not localised in adjacent normal breast tissue. In further analyses, the malignant and metastatic cases were combined and compared to adjacent normal breast cases. LYPD3 expression was decreased in low stage breast cancer lesions (stage I) (*p* = 0.028) and was observed in low grade lesions, but the latter did not reach significance. There was no significant association between LYPD3 and age, oestrogen receptor or HER2 status. 

### 2.5. LYPD3 Knockdown Significantly Reduces the Migration of MCF-7 Cells In Vitro

The levels of LYPD3 in the lysate and secretome of MDA-MB-468 cells was measured following adrenergic receptor stimulation and subsequent blocking with a β2-specific antagonist (Figure 3B). LYPD3 protein expression was increased in norepinephrine (*p* = 0.0375) and isoproterenol (*p* = 0.0234) stimulated cells with a subsequent decrease observed following treatment with an ADRβ2 antagonist (Figure 3C). LYPD3 mRNA expression levels were low in normal tissues (Figure 3D) and when the significantly associated proteins identified in the mass spectrometry analysis of MDA-MB-468 cell lysates and secretome were subjected to network inference and subsequently altered pathways were identified via the David annotation tool, it could be observed, as expected, that the most significant pathways were associated with cell proliferation, migration and adhesion (Figure 3E and Appendix A). The expression of LYPD3 in breast cancer cell lines varied with MCF-7 cells expressing the highest levels. This expression could be knocked down to 26.4% of its basal level using a LYPD3 specific shRNA (Figure 3F). MCF-7 cells were transfected with vehicle alone, shControl or shLYPD3 and stimulated with or without norepinephrine for 24 h to induce cell migration. Following norepinephrine treatment, cell migration significantly increased in MCF-7 vehicle cells (*p* = 0.0072) and shControl (0.0268) but not in the shLYPD3. The increase in migration observed in MCF-7 vehicle cells is more than originally observed (Figure 2B) and this could be due to two factors; a lower concentration of norepinephrine was used to stimulate the cells (100 nM vs. 10 µM) and the cells used in the knockdown studies were of a slightly lower passage number (passage 5 vs. passage 8). The difference in cell migration observed between the norepinephrine stimulated vehicle only cells and the shControl cells was not statistically significant (*p* = 0.1142).

In addition, shLYPD3 knockdown of norepinephrine stimulated MCF-7 cells caused a significant decrease in migration compared to vehicle alone cells (*p* = 0.0065) and shControl cells (*p* = 0.0016). Furthermore, in unstimulated cells there was a significant decrease in cell migration of shLYPD3 compared to shControl (0.0054) and vehicle treated cells (0.0188) (Figure 3G and Appendix A). These results indicate that cell migration may be influenced by LYPD3 acting via ABRβ signalling pathways.

### 2.6. Elevated Levels of LYPD3 mRNA Are Present in Malignant Disease Compared to Their Non-Malignant Counterpart in Several Cancers

Although previous studies have shown that LYPD3 is expressed in malignant tissues [20,21,22], a comprehensive screen of recent mRNA expression data has, to the best of our knowledge, not been performed. In our analysis we screened all cancers for the comparative mRNA expression levels between malignant and normal counterparts using GEPIA [23]. Our analysis revealed that in seven cancers (breast, cervical squamous cell carcinoma and endocervical adenocarcinoma, lung adenocarcinoma, lung squamous cell carcinoma, pancreatic adenocarcinoma, testicular germ cell tumours and thymoma) there is a significant (*p* < 0.001) over-expression of LYPD3 in the malignant tissue versus the normal counterpart (Figure 4). This suggests that LYPD3 could be a potential therapeutic target in multiple different cancers, and is not solely restricted to breast, oesophageal, pancreatic and lung carcinomas. 

## 3. Discussion

Early population studies reported an association between beta-blocker use and survival in patients with breast cancer [10,11,12]. The proposed physiological mechanism involved inhibition of β-adrenoceptor-activated cancer cell signalling pathways [8,19,23,24,25,26,27,28,29,30,31], resulting in a reduction of metastasis [9,32,33]. Specific mediators implicated downstream of this process have, to our knowledge, not been identified. Furthermore, the influence of adrenergic ligands on cell migration in vitro have been controversial [7,8,9]. We found that two of the non-stimulated triple-negative basal-type breast cancer models, MDA-MB-231 and MDA-MB-468, showed significantly raised levels of β2-adrenoceptor gene and protein expression compared to the oestrogen responsive cell line. Although a challenge with a non-selective ADRβ2 agonist confirmed functional adrenoceptors in all four cancer cell lines, the strongest response was seen in MDA-MB-231 and MDA-MB-468, as evidenced by increased cAMP. 

Although laboratory evidence supporting a clinical use for the beta-blocker propranolol in breast oncology is strong, it has not been universal, in that propranolol is reported by some to increase the proliferation of basal-type MDA-MB-231 breast cancer cells [34,35,36,37], but not by others [19,38]. We did not demonstrate a significant increase in survival in the breast cancer cells in response to a selective β2-adrenoceptor antagonist (ICI-118,551). A recent orthotopic mouse model of basal-type MDA-MB-231 breast cancer concurred that propranolol does not reduce primary tumour growth, but instead reduces the number and size of metastatic tumours [9]. In summary, beta-blockers appear not to be cytotoxic *per se*, but when combined with paclitaxel chemotherapy they appear to enhance its effectiveness, as demonstrated in a mouse basal-type (MDA-MB-231) breast cancer model [27]. 

Herein, the association between ADRβ2 expression and key characteristics involved in metastasis was assessed in four breast cancer models, including cell migration, adhesion and invasion. Migration assays confirmed that the agonist norepinephrine increased the migration of MDA-MB-468, BT-549 and MCF-7 cells, which was abrogated by the ADRβ2 antagonist ICI-118,551. In contrast, norepinephrine reduced the migration of MDA-MB-231 cells, and ICI-118,551 increased their migration, as has been reported by Carrie and Sebti [19]. However, when grown on collagen-1 or fibronectin protein-coated plastic or synthetic membranes coated with basement membrane protein, MDA-MB-231 cells showed a similar migratory behaviour to the other models (data not shown), thereby supporting the findings of others [8,39,40]. In addition, two basal-type cell lines (MDA-MB-231, MDA-MB-468) and the ER-positive MCF-7 cells exhibited increased norepinephrine-induced invasive behaviour that was abrogated by the ICI-118,551 antagonist. These findings suggest ADRβ2 mediated cellular pathways could be influenced by cell adhesion molecules [41], and may explain anomalous behaviour reported for the MDA-MB-231 cancer model when grown on different surfaces. This is further supported by the identification of proteins involved in cell adhesion, proliferation and migration following a mass spectrometry analysis of MDA-MB-468 cell lysates and secretome after ADRβ2 agonism/antagonism. The differences in migration observed between the MDA-MB-468 and MDA-MB-231 cells could, in part, be explained by differences in signalling arising because MDA-MB-468 cells are of epithelial origin (i.e., pre-epithelial-mesenchymal transition EMT), whilst MDA-MB-231 cells possess a mesenchymal-like phenotype (post-EMT). In MDA-MB-468 cells, norepinephrine could be signalling to initiate the acquisition of a more motile cell phenotype that would allow the cancer cells to spread, whereas in the MDA-MB-231 cells the presence of norepinephrine in the tissue microenvironment would signal that there is a favourable environment present for the cancer cells to successfully colonise and establish a secondary tumour/metastatic lesion [42,43,44,45]. Literature also demonstrates that catecholamines can both protect cancer cells from apoptosis [33] as well as providing an advantageous environment whereby secondary tumours often become established at catecholamine-producing tissues such as the brain and adrenal glands [46]. Furthermore, we hypothesise that the opposing influence of adrenergic stimulation on cell migration of MDA-MB-468 and MDA-MB-231 cell lines could be linked to ADR heterodimerisation. Lavoie et al. showed that heterodimerisation of ADRβ1 and ADRβ2 prevented agonist-induced internalisation of ADRβ2 [47] and in our study we demonstrated that MDA-MB-468 cells expressed higher levels of the ADRβ1 receptor than MDA-MB-231 cells. Furthermore, pharmacological diversity could be introduced via heterodimerisation leading to differential desensitisation, changes in cell surface receptor expression and alterations to the functionality of ADR subtypes [48,49,50,51,52,53].

Clearly there are many conflicting results in the literature for the two-dimensional migration of MDA-MB-231 cells and its variants [7,8,11,36,42]. There are a number of plausible reasons behind the differing results observed between the reports in the literature; including, the differing passage numbers of cells, different sources of the cell lines, the different methods and conditions used in performing the assays as well as time spans of the experiments. In addition, other reasons could be: (i) differing levels of β2-adrenoceptor signalling capacity in relation to the differing levels of β2 adrenoceptor expression and (ii) differing dose-dependent downstream signalling behaviours of the cell lines, including cAMP signalling. Not unsurprisingly, when the GO biological processes associated with the treatment of MDA-MB-231 cells with norepinephrine was investigated in our proteomic study, a number of genes associated with the following pathways were identified; proliferation, migration, adhesion, negative regulation of the apoptotic process and membrane organisation (Appendix A). Furthermore, Madden et al. has also shown that MDA-MB-231 cells can possess an impaired cAMP signalling, with cAMP levels remaining high after stimulation by forskolin, which could have profound effects and could possibly be a reason for the observed differences with this cell line [28]. In addition, Pon et al. demonstrated that the parental MDA-MB-231 cell line does not possess the feedforward Ca^2+^/cAMP loop and therefore has low levels of cAMP production and a delayed response to stress [25].

Furthermore, Kim et al. and Choy et al. used different variants of the parental MDA-MB-231 cell line, which could have profound influence on the in vitro behaviour of the cells in response to external stimulants such as isoproterenol [8,42]. Indeed, Pon et al. showed that the highly-metastatic variant of the MDA-MB-231 cell line, MDA-MB-231^HM^ are more responsive to β-adrenergic signalling than the parental MDA-MB-231 cell line and this is reflected in the limited cAMP signalling observed in the parental MDA-MB-231 cells compared to the high metastatic MDA-MB-231^HM^ variant [25]. The two-dimensional migratory activity of MDA-MB-231 cells in response to isoproterenol has also been assessed on different extracellular matrix (ECM) protein surfaces where again, opposing effects were observed, even on the same surface, adding to the conflicting results of this cell line in the literature [26,43].

We report that the strongest association between high ADRβ2 expression and cell behaviours indicative of tumour aggression exists in two triple-negative basal-type breast cancer models (MDA-MB-231, MDA-MB-468). Moreover, we have identified distinctly different protein profiles between the two models in response to ADRβ2 activation, with differential expression of the pro-metastasis protein LYPD3 observed in MDA-MB-468 cells following ADR stimulation and/or ADRβ2 antagonism/inverse agonism. Stimulation of MDA-MB-468 cells with norepinephrine or isoproterenol increased the levels of LYPD3 within the cell lysate, and was also secreted in the media. Treatment with the antagonist ICI-118,551 reduced LYPD3 expression to a lower level than that observed following norepinephrine/isoproterenol stimulation both in the lysate and secreted into the media. LYPD3 is a glycosyl-phosphatidyl-inositol (GPI) anchored glycoprotein whose expression, we have confirmed, is highly restricted in normal tissues [54,55]. Studies have also demonstrated a strong association with a poor prognosis [20,22,56,57,58] and even though its expression has been observed in breast cancer both in this study and in other published studies [21,59], its influence on tumourigenesis is yet to be elucidated although our patient tumour microarray (TMA) data suggests that LYPD3 is exclusively expressed in primary breast cancer and metastatic cases, with no expression observed in normal breast tissue. Furthermore, our in silico analysis has revealed that LYPD3 may be a therapeutic target in multiple cancer types, some of which have not been reported in the literature (testicular germ cell tumours and thymoma). Interestingly, upregulation of LYPD3 has been observed following cellular stress [60], however, this is the first study showing that treatment of breast cancer cells with the endogenous stress hormone norepinephrine, can also lead to elevated LYPD3 levels.

In MDA-MB-468 cells, treatment with the non-selective ADRβ agonist, isoproterenol, unregulated LYPD3, whereas ICI-118,551, a selective ADRβ2 antagonist, reduced norepinephrine-induced LYPD3 expression and, when administered alone, reduced LYPD3 levels to that below the basal expression. This would suggest, through a mechanism yet to be elucidated, that LYPD3 is regulated via the ADRβ2 signalling pathway. In oesophageal cancer, LYPD3 is regulated via CREB (cAMP response element binding protein) transcription co-activator signalling [61], and therefore, it is postulated that norepinephrine regulates LYPD3 through an ADRβ2/cAMP/PKA/CREB/LYPD3 effector pathway [45]. In this pathway a conformational change in ADRβ2 would be induced following binding of norepinephrine, mediating activation of the G_sα_ protein subunit. Following activation, G_sα_ can then stimulate the adenylyl cyclase-induced conversion of adenosine triphosphate (ATP) into cAMP, resulting in protein kinase A (PKA) activation. Downstream this would enable phosphorylation of CREB via PKA, thereby inducing the transcriptional upregulation of LYPD3, and hence, increasing cell migration [45,61]. In this study we have demonstrated the importance of LYPD3 in breast cancer cell migration by successfully knocking down the expression of LYPD3 using LYPD3-specific shRNA. We found that cell migration, measured using transwell migration assays, was significantly reduced following shLYPD3 compared to shControl.

In the secretome of MDA-MB-468 cells, increased levels of LYPD3 and decreased levels of LAMC1 (Laminin Subunit Gamma 1) were also observed following treatment with norepinephrine and isoproterenol. This is an interesting observation because it is thought that LYPD3 can be cleaved from the cell surface by ADAM Metallopeptidase Domain (ADAM)-10 and -17 and this shedding can be induced by hypoxia [62]. To speculate, once LYPD3 has been shed, it may still be able to bind laminin, via associating with the α6β4 integrin and matrix metallopeptidase-14 (MMP-14) and contribute towards its fragmentation and the observed decrease in LAMC1 that we observed in our proteomic analysis [60,63]. Alternatively, like its structural homologue urokinase-type plasminogen activator receptor (uPAR), shed-LYPD3 could function as a chemoattractant [64]. 

In MDA-MB-231 cells, LYPD3 was not detected in the library of proteins generated from either the lysate or the secretome and this observation was confirmed by gene expression analysis of LYPD3 in a range of breast cell lines illustrating that MDA-MB-231 cells do not express LYPD3 at the mRNA level. LYPD3 was expressed at negligible levels in the BT-549, which is also a cell line of mesenchymal origin. Higher levels of expression of LYPD3 were observed in the epithelial-derived cell lines suggesting that LYPD3 expression may correlate with EMT. Furthermore, Harner-Foreman et al. published a spontaneous model of prostate cancer [65] and unpublished mass spectrometry proteomic profiling data from this study revealed that LYPD3 was downregulated post-EMT. This is further supported by observations made by Oshiro et al., where significant associations were made between LYPD3 and EMT in both colorectal cancer cell lines and in clinical samples [66]. 

## 4. Materials and Methods

### 4.1. Cell Lines

The following breast cancer cell lines were purchased from the American Tissue Culture Collection (ATCC): BT-549 (HTB-122™), ductal carcinoma; MCF-7 (HTB-22™), adenocarcinoma; MDA-MB-231 (HTB-26™), adenocarcinoma; MDA-MB-468 (HTB-132™), adenocarcinoma (Atcc.org, 2013). Breast cancer cell lines were cultured in DMEM (Dulbecco’s Modified Eagle Medium) (Lonza, Slough, UK) containing 10% *v*/*v* FCS (foetal calf serum) (GE Healthcare Life Sciences, Buckinghamshire, UK). All cells were grown at 37 °C, in a humidified atmosphere with 5% *v*/*v* CO_2_. After washing in PBS (phosphate-buffered saline), breast cancer cells were harvested using Trypsin and EDTA (ethylenediaminetetraacetic acid)l (Lonza, Slough, UK).

### 4.2. RNA Extraction, cDNA Synthesis and qRT-PCR

RNA was extracted using an RNeasy mini kit (QIAGEN, Manchester, UK), then quantified on a NanoDrop^TM^ 8000 Spectrophotometer. cDNA synthesis was performed using MMLV-reverse transcriptase (Promega, Southampton, UK) and oligo-dT primers (Promega, Southampton, UK). In brief, 1 μL oligo(dT)15 primer (Promega, Southampton, UK) was annealed to 2 μg of RNA. After being denatured for 5 min at 70 °C, a master mix solution was added, which contained: 0.7 μL Recombinant RNasin^®^ Ribonuclease Inhibitor (Promega, Southampton, UK); 1 μL Moloney-Murine Leukemia Virus (M-MLV) reverse transcriptase (Promega, Southampton, UK); 5 μL M-MLV 5× reaction buffer (Promega, Southampton, UK); 1 μL Deoxynucleotide Triphosphate Solution Mix (Sigma, Dorset, UK) and 2.3 μL nanopure water. The samples were incubated for 60 min at 37 °C and then heated for 5 min at 95 °C. cDNA was then stored at −20 °C.

Semi-quantitative real-time PCR was performed using SYBR^®^ Green (Bio-Rad, Watford, UK) chemistry and gene-specific primers (MWG Eurofins, Ebersberg, Germany) (Appendix A) on a Rotor-Gene 6000 real-time PCR cycler (QIAGEN, Manchester, UK). PCR reactions were performed in 0.1 mL strip tubes containing a 12.5 μL mixture of: SYBR^®^ Green (Bio-Rad, Watford, UK); nanopure water; gene-specific primers (MWG Eurofins, Ebersberg, Germany) at a concentration of 5 pmol; and either 0.5 μL of cDNA or nanopure water (control). The following cycling conditions were used (35–40 cycles): initial 5 min at 95 °C for enzyme activation, followed by denaturation at 95 °C for 10 s, annealing for 15 s at the primer–specific Tm (Appendix A), and extension at 72 °C for 20 s. Following each PCR the melt curves were examined prior to data analysis. Using the Rotor-Gene Q Software and a threshold of 0.08, each transcripts Ct (cycle threshold) value was determined in triplicate. The relative expression of each target gene was then semi-quantified using the 2^−ΔΔCT^ method.

### 4.3. Flow Cytometry

Unconjugated antibodies were fluorescently labelled using Lightning-Link^®^ (Innova Biosciences, Cambridge, UK), according to the manufacturer’s protocol: ADRα1B antibody with PE-Cy7™; ADRα1D antibody with APC (Allophycocyanin) and β2-adrenoceptor antibody with RPE (R-Phycoerythrin). Harvested cells (1 × 10^5^) were washed, pelleted and re-suspended in medium. Cells were then treated with an Fc receptor blocking reagent (Miltenyi Biotec, Bergisch Gladbach, Germany) diluted in PBS (10 min, 4 °C). Cells were incubated in the dark (30 min, 4 °C) with conjugated ADR antibodies (at a pre-optimised concentration) (Appendix A) and viable cells were identified using LIVE/DEAD™ fixable violet dead cell stain (Thermo Fisher Scientific, Leicester, UK). Cells (minimum 10,000) were analysed by flow cytometry (Beckman Coulter Gallios™, Kaluza™ software).

### 4.4. cAMP Signalling

Cells were treated with the non-selective β-agonist isoproterenol (1 µM, 10 min) and cAMP production measured using the cAMP Parameter^TM^ ELISA kit (R&D Systems, Minneapolis, MN, USA). IBMX was incorporated to prevent cAMP degradation and phosphodiesterase activity on cAMP production. 

### 4.5. Cellular Survival

Cells were seeded at 5000 cells/well in 100 µL of advanced DMEM containing 2% FCS and 4 mM L-glutamine. The cells were incubated at 37 °C, 5% CO_2_ in a humidified atmosphere for 24 h. After 24 h the media was carefully removed and the cells washed once with PBS. Fresh serum-free advanced DMEM was added to each well and the cells serum-starved for 24 h. After 24 h, the media was removed and the cells washed with PBS. 100 µL of advanced DMEM containing 2% FCS and 4 mM L-glutamine was adding to the wells with or without the addition of ICI-118,551 (β2-ADR selective antagonist, Sigma, Dorset, UK) (concentration range 1 pM–10 µM). Cells were treated with antagonists for 30 min prior to the addition of 10 µM norepinephrine (Sigma, Dorset, UK) for 72 h at 37 °C in a humidified atmosphere containing 5% CO_2_. After 72 h, 100 μL of 2× detection reagent was added to each well and the plate incubated for 60 min. The plate was then read using a Tecan Ultra fluorescent plate reader (Tecan Ultra, Mannedorf, Switzerland; excitation 485 nm, emission 535 nm). 

### 4.6. Adhesion Assays

Cells were resuspended in 10 mL of serum-free media at 200,000 cells/mL in a falcon tube and were incubated for 30 min to allow recovery from detachment. 10 μM ICI-118,551 was added to the cells for 30 min prior to the addition of norepinephrine (at pre-optimised concentrations). 50 μL of serum-free media containing norepinephrine or media alone was added to each well of the 96-well plate and the plate was incubated for 30 min. After 30 min, 10,000 cells/well of cell suspension was added to each well of the 96-well plate [96-well plates pre-coated with human fibronectin (1.0 μg/well), human vitronectin (0.5 μg/well) (R&D Systems, Minneapolis, MN, USA), or collagen I (3 mg/mL) (Thermo Fisher Scientific, Leicester, UK)]. Untreated cells were added to wells containing 50 μL of pre-incubated media containing DMSO. Treated cells were added to wells containing 50 μL of pre-incubated media containing 10 µM norepinephrine. The plate was then incubated for 3 h at 37 °C in a humidified atmosphere containing 5% CO_2_. The media was removed from each well and the cells washed carefully twice with PBS. The final PBS wash was aspirated, the cells fixed with 4% paraformaldehyde in PBS and the plate incubated at room temperature for 20 min. The wells were washed twice with PBS. Cells were stained by adding 50 μL of crystal violet/cell stain solution and incubated at room temperature for 15 min. Wells were washed twice with deionised water and the last wash aspirated. The wells were allowed to fully dry at room temperature. Once dry, the wells of the plates were scanned using a C.T.L. ELISPOT plate reader and the number of remaining cells counted using ImmunoSpot^®^ software (ImmunoSpot, Bonn, Germany). 

### 4.7. Cultrex^®^ Cell Migration Assay

Cells were serum-starved for 24 h prior to performing the assay. 1 × 10^6^ cells/mL in serum-free media (MDA-MB-231, BT-549) or 0.5% FCS containing media (MDA-MB-468, MCF-7) were added to a 1.5 mL microtube along with 10 μM ICI-118,551. 50 μL/well of cell suspension (50,000 cells) with or without antagonists was added to the top chamber of the plate followed by the addition of 150 μL of DMEM containing 10% FCS to the bottom chamber of each well. The cells were incubated for 30 min prior to the addition of norepinephrine (10 μM). The plate was incubated for 24 h at 37 °C in a humidified atmosphere containing 5% CO_2_. After incubation, the top chamber was inverted and carefully shaken to remove the culture medium and transferred to the black receiver plate. Each well of the top chamber was washed with 100 μL of warm 1× wash buffer and the top chamber inverted and carefully shaken to remove excess wash buffer and placed back into black receiver plate. To the bottom chamber of each well is added 100 μL of 1000× cell dissociation solution/ Calcein AM solution and the plate incubated at 37 °C in a humidified atmosphere containing 5% CO_2_ for 60 min. The plate was read on a fluorescent plate reader at 485 nm excitation, 520 nm emission (Tecan Ultra, Mannedorf, Switzerland). 

### 4.8. CultreCoat^®^ Medium BME Cell Invasion Assay

Cells were serum-starved for 24 h prior to performing the assay. 1 × 10^6^ cells/mL in serum-free media (MDA-MB-231, BT-549) or 0.5% FCS containing media (MDA-MB-468, MCF-7) were added to a 1.5 mL microtube along with 10 μM ICI-118,551. 25 μL/well of cell suspension (25,000 cells) were added to the top chamber of the plate followed by the addition of 150 μL of DMEM containing 10% FCS to the bottom chamber of each well. The cells were incubated for 30 min prior to the addition of 10 μM norepinephrine. The plate was incubated for 24 h at 37 °C in a humidified atmosphere containing 5% CO_2_. After incubation the top chamber was washed with 100 μL of warm 1× wash buffer and placed into a black receiver plate. To the bottom chamber of each well 100 μL of 1000× cell dissociation solution/Calcein AM solution is added and the plate incubated at 37 °C in a humidified atmosphere containing 5% CO_2_ for 60 min to fluorescently label and dissociate cells from the membrane. After 60 min, the top chamber was removed and fluorescence was measured using a plate reader (Tecan Ultra, Mannedorf, Switzerland; 485 nm excitation, 520 nm emission). 

### 4.9. Proteomic Analysis

MDA-MB-231 (1.3 × 10^6^) and MDA-MB-468 (1.2 × 10^6^) cells were seeded and grown to 90% confluency. On the day of treatment 1 mM stocks of norepinephrine, isoproterenol and ICI-118,551 were made up in serum free DMEM and passed through a 0.22 µm filter prior to serial dilution. Media was removed from each flask, and cells were washed three times with PBS. Untreated cells received 25 mL of serum free DMEM, and treated cells received 25 mL of DMEM containing 10 μM of the appropriate treatment condition. After 30 min incubation 10 µM norepinephrine or 10 µM isoproterenol was added into the relevant flasks. Flasks were then incubated at 37 °C, 5% CO_2_ for 24 h. Twenty four hours after treatment secretome samples were obtained by removing the media and centrifuging (300× *g*, 5 min), filtered (0.22 µm) and concentrated using Amicon Ultra-15 Centrifugal Filter Units (Merck, Kenilworth, NJ, USA). Cell lysates were prepared using 9.5 M urea (Melford, Stowmarket, UK)/2% *v*/*v* dithiothreitol (Melford, Stowmarket, UK)/1% *v*/*v* n-octyl-beta-glycopyranoside (Apollo Scientific Limited, Stockport, UK) containing protease inhibitor (Sigma, Dorset, UK). The lysates were harvested, chilled on ice for 5 min, sonicated for 5 min and this process was repeated three times before centrifugation for 10 min at 12,000× *g* and storage at −80 °C. 

Cell lysate (100 µg) and secretome (100 µg) were diluted in 50 mM tri-ethyl ammonium bicarbonate (TEAB) before reduction (5 mM DTT at 56 °C for 20 min) and alkylation (15 mM iodoacetamide at room temperature for 15 min) and then digested for 16 h using Trypsin/Lys-C (Promega, Southampton, UK) at 37 °C at a 20:1 protein:protease ratio (*w/w*) in a thermomixer (650 rpm) [67]. Next, samples were cleaned up using HyperSep C_18_ cartridges (Thermo Scientific, Leicester, UK) for solid phase extraction. A vacuum concentrator was then used to concentrate the samples before resuspension in 5% acetonitrile + 0.1% formic acid and subsequent analysis of the peptides on an AB Sciex TripleTOF 5600+ MS/MS instrument in both SWATH (Sequential Window Acquisition of All Theoretical Mass Spectra) and IDA (information dependent acquisition) acquisition modes (Sciex, Framingham, MA, USA). 

### 4.10. Mass Spectrometry

Each sample was analysed on a Sciex TripleTOF 5600+ mass spectrometer coupled in line with an Eksigent ekspert nano LC 425 system running in micro flow (5 µL/min) mobile phase B (100% acetonitrile + 0.1% formic acid) over mobile phase A (0.1% formic acid). In brief, 8 µg of sample was injected and trapped onto a YMC Triart-C_18_ pre-column (5 mm, 3 µm, 300 µm ID) at a flow rate of 10 µL min for 2 min. The sample was then eluted off the pre-column and onto a YMC Triart-C_18_ analytical column (15 cm, 2 µm, 300 µm ID) that was in line with the Sciex TripleTOF 5600+ Duospray Source using a 50 µm electrode in positive mode, +5500V. The following linear gradients were used: for SWATH, mobile phase B increasing from 3% to 30% over 38 min, 30% to 40% over 5 min, 40% to 80% over 2 min for wash and re-equilibration (total run time 57 min). For IDA, mobile phase B increasing from 3% to 30% over 68 min, 30% to 40% over 5 min, 40% to 80% for column wash and re-equilibration over 2 min (total run time 87 min). Data independent acquisition was performed using 100 variable SWATH windows (optimised on sample type) (TOFMS *m*/*z* 400-1250) 25 ms accumulation time; 2.6 s cycle and IDA with a top 30 ion fragmentation (TOFMS *m*/*z* 400-1250) followed by 15 s exclusion using rolling collision energy, 50 ms accumulation time; 1.8 s cycle. 

Spectral library generation, alignment and fold change analysis were performed as described previously [68]. In brief, IDA data were searched using ProteinPilot 5.0 (iodoacetamide alkylation, biological modifications emphasised in a thorough search) against the Swiss-Prot human database (June 2017). The Sciex OneOmics software was used to analyse the SWATH data following extraction against the locally generated library with the following parameters; 12 peptides per protein, six transitions per peptide, XIC width 30 ppm, 6 min retention time window.

### 4.11. LYPD3 Protein Expression

Mass spectrometry and curated protein profiling of the two basal-type cell lines (MDA-MB-231 and MDA-MB-468) identified increased LYPD3 expression, a pro-metastasis protein. LYPD3 protein expression in breast cancer tissue was confirmed by immunostaining three wax-embedded TMA slides (BC081120b, BC10010d, BR1201) comprising 260 cases of invasive ductal adenocarcinoma, 50 cases of metastatic adenocarcinoma and 10 cases of normal breast tissue (U.S Biomax, Rockville, MD, USA). 

Sections were immunostained using a monoclonal rabbit anti-human LYPD3 antibody (Appendix A) on a BenchMark ULTRA stainer (Ventana Medical System, Inc, Oro Valley, AZ, USA) with ultraView Detection Kit (Ventana Medical, Oro Valley, AZ, USA). 

The intensity of LYPD3 immunostaining was microscopically assessed for cytoplasmic staining using a five-point scoring technique, where a score of 0 represented nil staining; a score of 1: weak; 2: moderate; 3: strong; 4: strong cytoplasmic staining with additional cell membrane staining. 

The association between LYPD3 expression and clinical variables in malignant adenocarcinoma, metastasis and adjacent normal breast tissue was statistically tested using SPSS (Version 24, IBM, UK). Immunohistochemistry scores were dichotomously categorised (0, 1 = negative; 2, 3, 4 = positive) for Chi-square tests. Significance levels were *p* = 0.05 or less. 

### 4.12. Western Blot

Total cell lysates were used from the proteomic analysis. In brief, 50 µg of total protein from each sample was prepared with sample reducing buffer (60 mM Tris-HCl (pH 6.8), 2% SDS (sodium dodecyl sulphate), 10% glycerol and 0.01% bromophenol blue) at a ratio of 1:3 sample vs. reducing buffer. The sample was resolved on an SDS gel (10% resolving gel, 5% stacking gel) with Tris/glycine/SDS gel running buffer (Geneflow) at a constant voltage of 150 V. After separation, samples were transferred onto nitrocellulose membranes using Tris/glycine/methanol transfer buffer at a constant current of 180 mA for 75 min at 4 °C. Membranes were blocked in 10% Marvel™ dried skimmed milk powder for 1 h before being probed with rabbit anti-LYPD3 antibody (1:1000, ab151709, Abcam, Cambridge, UK), rabbit anti-beta 2 adrenergic receptor antibody (1:1000, ab182136, Abcam, Cambridge, UK) and rabbit anti-beta actin antibody (1:5000, ab8227, Abcam, Cambridge, UK) overnight at 4 °C. The membranes were then washed and goat anti-rabbit IgG HRP-linked antibody was added (1:1000, Cell Signalling Technology, London, UK). Membranes were washed and exposed to the Clarity Western ECL Substrate (1:1) and imaged using a Syngene G:Box and Genesys v1.5.4.0 software (Syngene, Cambridge, UK).

### 4.13. Generation of LYPD3 Knockdown Cell Line

Lentiviral shRNA plasmids and packaging mix (SHP001) were purchased from Sigma (Dorset, UK): shLYPD3 (catalogue number: SHC204) and shControl (catalogue number: SHCLNV-NM_133743). MCF-7 cells (2.5 × 10^5^) were transfected with shRNAs and hexadimethrine bromide (Sigma, H9268) at a final concentration of 8 µg/mL for 18 h. Resistant colonies were selected using media containing 2 µg/mL puromycin.

### 4.14. In Silico Gene Expression Profiling

LYPD3 gene expression profiling was performed using patient gene expression profiles generated through The Cancer Genome Atlas (TCGA) [69] and the Genotype-Tissue Expression (GTEx) project [70] and data was assessed via the Gene Expression Profiling Interactive Analysis (GEPIA) [23].

## 5. Conclusions

For the first time, this study postulates that ADRβ2 signalling can regulate LYPD3. Furthermore, by knocking down LYPD3 expression in the MCF-7 breast cancer cell line we demonstrated that LYPD3 supports both basal and norepinephrine-induced cell migration. Thus, its level of expression could be an important indicator of breast tumour progression and this was supported by our findings in breast cancer tissue cohorts. This highlights LYPD3 as a promising therapeutic target but also supports the development of novel beta-blocker compounds that could be used prophylactically to try and reduce metastatic spread. 

## Figures and Tables

**Figure 1 biology-09-00039-f001:**
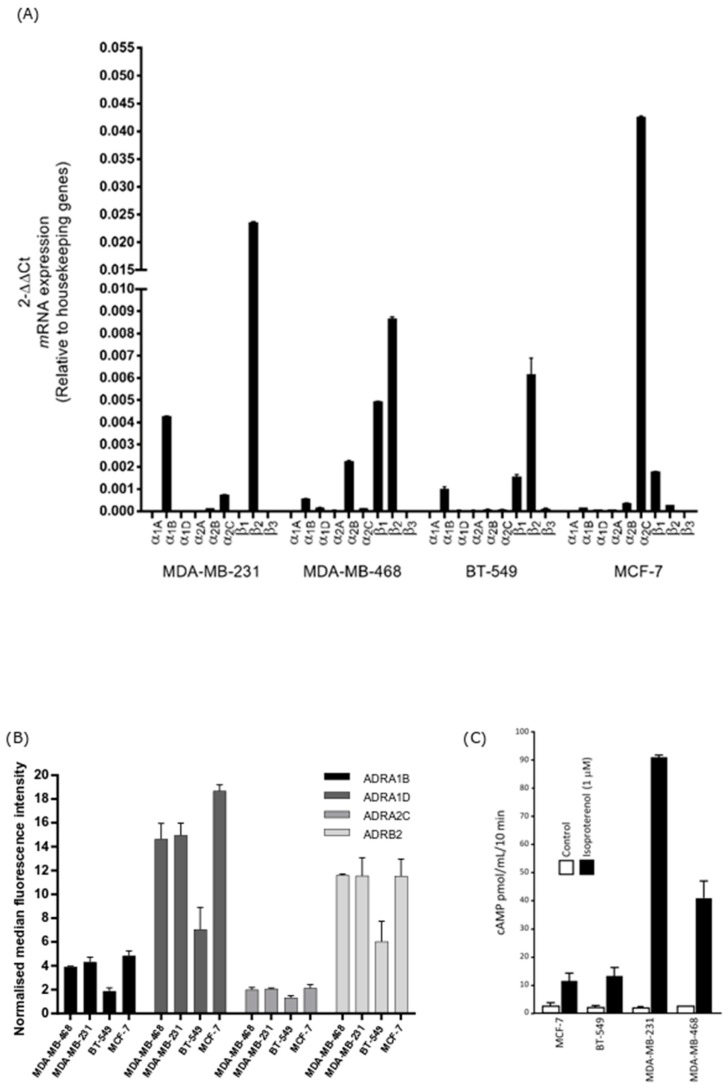
Adrenoceptor (ADR) expression on breast cancer cells and measurement of cAMP (cyclic adenosine monophosphate) levels following adrenergic stimulation. (**A**) Relative expression of ADR mRNA in breast cancer cell lines was quantified by qRT-PCR (quantitative reverse transcription polymerase chain reaction) and relative expression (2^−ΔΔCT^) was determined by normalisation to housekeeping genes. β2-adrenoceptor gene expression was strongest in the basal-type cell lines MDA-MB-231 and MDA-MB-468. (**B**) Expression of the adrenoceptors on unpermeabilised breast cancer cells was assessed by measuring median fluorescence intensity (MFI) using flow cytometry. Membranous β2-adrenoceptor protein expression was highest in the basal-type cell line MDA-MB-468, followed by MDA-MB-231 > MCF-7 > BT-549. (**C**) Isoproterenol (β-agonist) stimulated cAMP accumulation in breast cancer cell lines. Cells were treated in the presence of IBMX to prevent cAMP degradation. cAMP production was highest in the order MDA-MB-231 > MDA-MB-468 > BT-549 > MCF-7. All assays were performed in triplicate (*n* = 3). Results shown are the mean ± standard deviation.

**Figure 2 biology-09-00039-f002:**
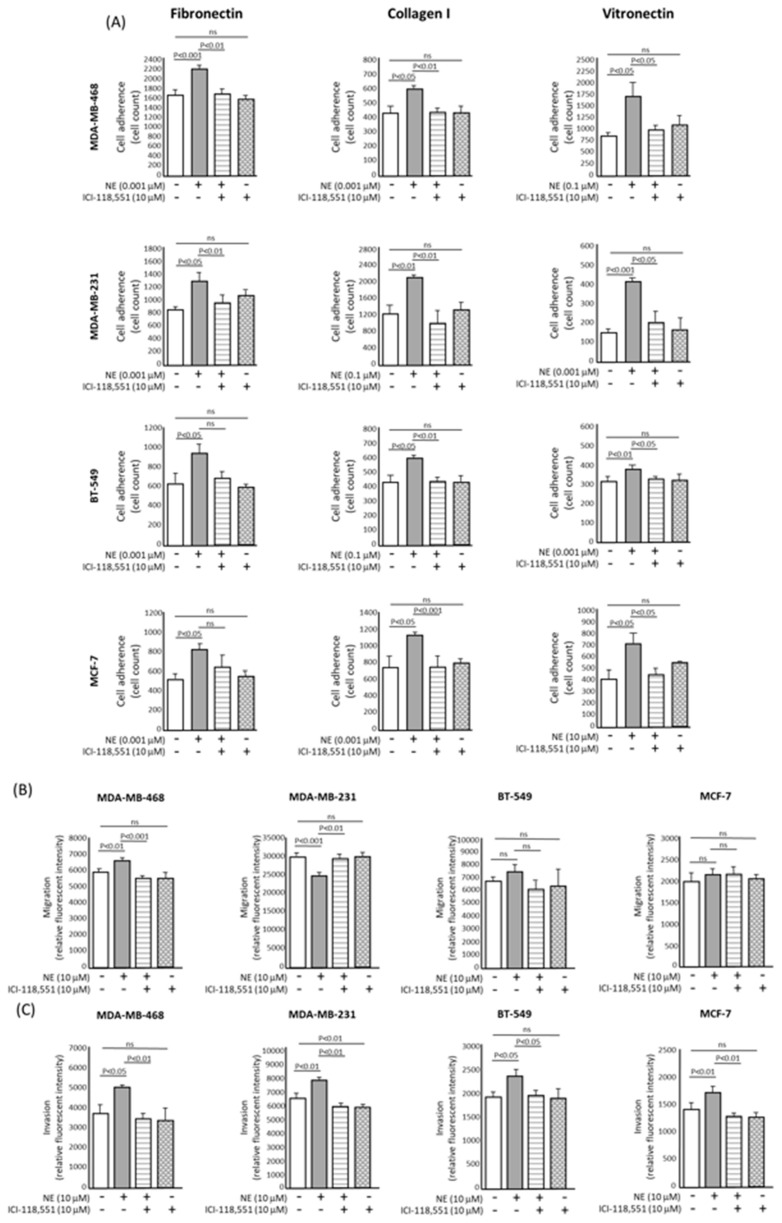
Effect of norepinephrine and the selective β2-adrenoceptor antagonist ICI-118,551 on breast cancer cell adhesion, migration and invasion in vitro. (**A**) β2-adrenoceptor blockade using ICI-118,551 inhibits norepinephrine-induced adhesion of breast cancer cells to fibronectin and collagen I in vitro. Quantitative analysis of the number of cells adhered to wells coated with 10 μg/mL of fibronectin, collagen I or vitronectin after 3 h of incubation. Simultaneous treatment with norepinephrine and the selective antagonist ICI-118,551 compound reduced the adhesion in all four cell lines grown on collagen I and vitronectin coated surfaces (*p* < 0.05). Only the basal-type MDA-MB-468 and MDA-MB-231 cell lines showed a reduction in cell adhesion when grown on fibronectin (*p* < 0.01). (**B**) Quantitative analysis of the influence of ICI-118,551 treatment on the transwell migration of the breast cancer cell lines. Norepinephrine challenge caused a significant increase in the migration of MDA-MB-468 cells (*p* < 0.05) and this increase was completely reversed by the concomitant administration of ICI-118,551 antagonist (*p* < 0.05). (**C**) Quantitative analysis of the influence of ICI-118,551 treatment on the invasion of the breast cancer cell line through a basement membrane epithelium (BME) coated membrane. Treatment with the ICI-118,551 antagonist significantly decreased the invasive capacity of MDA-MB-468, MDA-MB-231 and MCF-7 cell lines following norepinephrine challenge (*p* < 0.05). All assays were performed in triplicate (*n* = 3). Results shown are the mean ± standard deviation. Statistical analysis was performed using a one-way ANOVA (analysis of variance) test (Dunnett’s multiple comparison test).

**Figure 3 biology-09-00039-f003:**
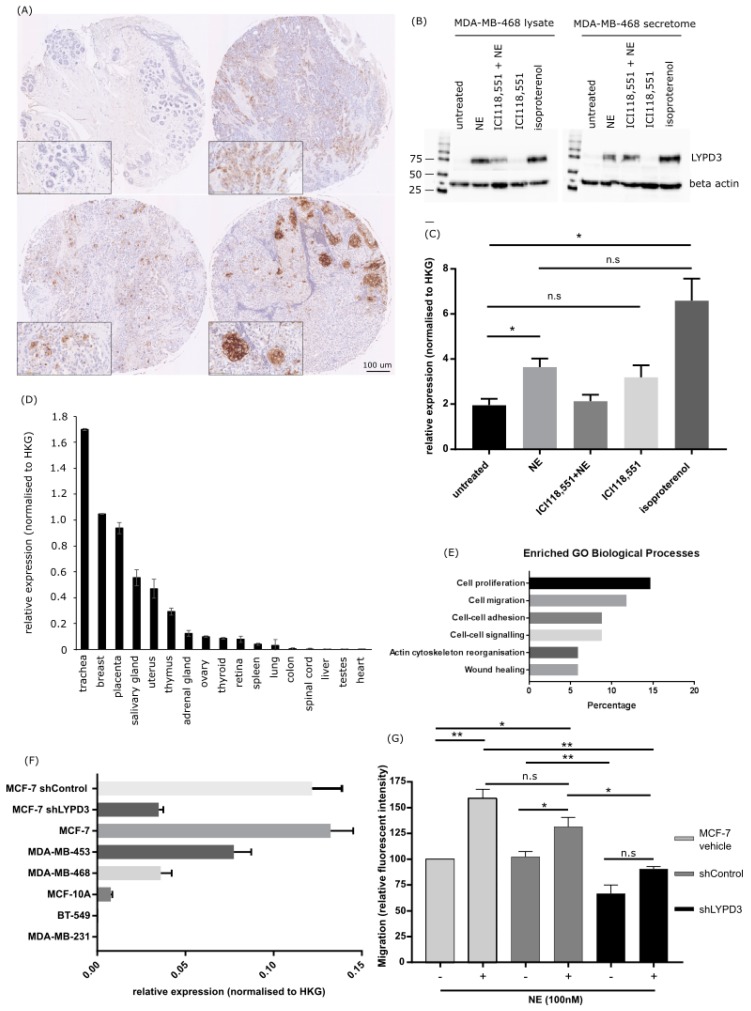
LYPD3 expression in breast patient TMAs (tumour micro arrays), breast cancer cell lines and a normal tissue RNA panel and the effects of LYPD3 gene knockdown. (**A**) IHC (immunohistochemistry) staining of LYPD3 in (top left) normal breast, (top right) invasive ductal carcinoma (TN, grade 3, stage I), (bottom left) invasive ductal carcinoma (TN, grade 3, stage IIA), (bottom right) lymph nodes metastases from patients with invasive ductal carcinoma. (**B**) Western blot of LYPD3 expression in 50 µg protein extracted from the cell lysate (left hand panel) or secretome (right hand panel) of MDA-MB-468 cells following treatment demonstrated an upregulation of LYPD3 protein following treatment with norepinephrine or isoproterenol alone and also following concomitant treatment with ICI118,551. (**C**) LYPD3 gene expression level in MDA-MB-468 cells following treatment demonstrates a significant upregulation of LYPD3 in norepinephrine and isoproterenol stimulated MDA-MB-468 cells. Results shown are the mean ± standard deviation. Statistical analysis was performed using an un-paired Student’s T-test. * *p* < 0.05, ** *p* < 0.01. (**D**) LYPD3 gene expression levels (relative to the house keeping gene—HKG) in a panel of normal human tissues illustrates low levels of expression in the majority of essential tissues tested (lung, spinal cord, liver, heart) (**E**) Enriched GO biological processes (as defined by David annotation tool) of differentially expressed proteins identified via the mass spectrometry analysis within the top 50 strongest interactions as defined by network inference analysis. The normalised protein expression values of each identified protein (cut-off above 50% confidence) within the untreated and norepinephrine treated cohorts (both lysate and secretome) were subjected to a network inference analysis. In the network, the 50 strongest interactions (based on their absolute values) between any of the proteins (defined by their strength of interaction) were selected and submitted to the David annotation tool and as observed the top pathways identified were those associated with cell proliferation, migration, adhesion and cell signalling. (**F**) LYPD3 gene expression levels in a panel of breast cell lines (relative to the house keeping gene—HKG) showing that the more mesenchymal cell lines (MDA-MB-231 and BT-549) express negligible levels of LYPD3 compared to the more epithelial cell lines (MDA-MB-468, MDA-MB-453) with MCF-7 cells demonstrating the highest level of expression. Subsequently, the level of LYPD3 expression in MCF-7 cells could be reduced by the incorporation of LYPD3 specific shRNA to 26.4% of its original levels. (G) The effects of LYPD3 gene knockdown on the migration of MCF-7 breast cancer cells showed a significant reduction in the migration of MCF-7 vehicle cells following norepinephrine stimulation (*p* = 0.0065). There was also a reduction in the migration of MCF-7 cells following LYPD3 shRNA knockdown compared to shControl (*p* = 0.0054). All assays were performed in triplicate (*n* = 3). Results shown are the mean ± standard deviation. Statistical analysis was performed using an ANOVA test (Dunnett’s multiple comparison test). * *p* < 0.05, ** *p* < 0.01.

**Figure 4 biology-09-00039-f004:**
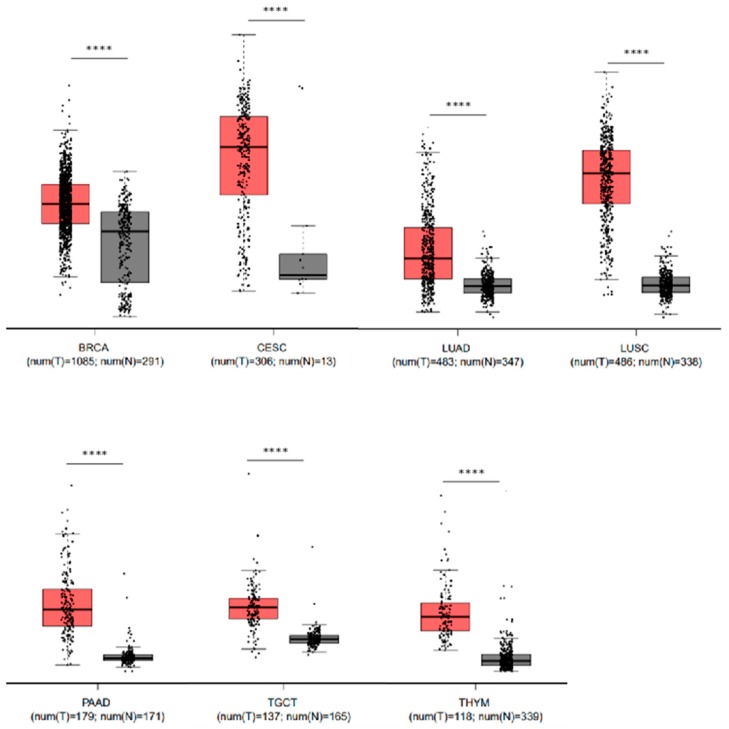
LYPD3 mRNA expression is significantly (*p* < 0.001) upregulated in malignant cancer (T) versus the normal tissue (N) counterpart. Box and whisker plots illustrating the mRNA expression levels of LYPD3 in the following cancer versus normal tissue: BRCA—breast cancer, CESC—cervical squamous cell carcinoma and endocervical adenocarcinoma, LUAD—lung adenocarcinoma, LUSC—lung squamous cell carcinoma, PAAD—pancreatic adenocarcinoma, TGCT—testicular germ cell tumours and THYM—thymoma. Cancer are coloured red whilst normal are coloured grey. Each point represents an individual data point and the number of data points in each analysis is stated under the *x*-axis. The log2 fold change cut-off is 1 and differential analysis is by one-way ANOVA, using disease state as the variable for calculating differential expression.

**Table 1 biology-09-00039-t001:** Differentially expressed proteins identified using SWATH (Sequential Window Acquisition of All Theoretical Mass Spectra) MS (mass spectrometry) analysis of MDA-MB-468 cell lysate and secretome following adrenoceptor agonism/antagonism.

	Lysate			Secretome		
Treatment (vs UT)	Protein	Swiss-Prot ID	Log2 Fold Change	Protein	SwissProt ID	Log2 Fold Change
NE	LYPD3	O95274	2.235	IF5A1	P63241	2.181
	CRYAB	P02511	1.342	H2BFS	P57053	2.066
	NDRG1	Q92597	1.021	LYPD3	O95274	1.577
	CASPE	P31944	1.015	H4	P62805	1.489
	AATM	P10809	−1.010	STC1	P52823	1.077
	CH60	P10809	−1.141	LIF	P15018	−2.090
	HINT2	Q9BX68	−1.206			
	MDHM	P40926	−1.315			
	ECH1	Q13011	−1.431			
	H14	P10412	−1.515			
	ODO4	P36957	−1.520			
	H12	P16403	−1.574			
ISO	IF172	Q9UG01	2.684	DPY30	Q9C005	1.320
	LYPD3	O95274	2.564	ADML	P35318	1.142
	CRYAB	P02511	1.413	H4	P62805	1.078
				ANXA1	P04083	1.065
				CTGF	P29279	−1.151
				UBA3	Q8TBC4	−1.995
				LAMC1	P11047	−3.499
ICI 118,551	S10-A7	P31151	1.271	NDKA	P15531	3.784
				IF5A1	P63241	2.252
				S10A6	P06703	1.651
				ANXA2	P07355	1.311
				DPY30	Q9C005	1.286
				EDF1	O60869	−2.432
ICI 118,551 + NE	KRT36	O76013	2.160	CRK	P46108	2.591
	CRYAB	P62750	1.776	NUCL	P19338	1.947
	NDRG1	P02768	−1.067	HS90A	P07900	1.887
				APOA1	P02647	1.769
				ANXA2	P07355	1.694
				CUTA	O60888	1.642
				STC1	P52823	1.474
				FETUA	P02765	1.418
				LYPD3	O95274	1.308
				H4	P62805	1.140
				S100P	P25815	1.139
				SFRP1	Q8N474	1.136
				TSP1	P07996	−1.280
				EDF1	O60869	−4.362
				BSSP4	Q0GZN4	−4.870

Log2 fold change represents the increase/decrease in protein expression relative to untreated samples (UT). Proteins with a Log2 fold change of <1.0 or >1.0 and OneOmics confidence of ≥60% are shown. Proteins with single peptides are excluded. Data is from six biological replicates. NE—norepinephrine, ISO—isoproterenol, UT—untreated.

**Table 2 biology-09-00039-t002:** Differentially expressed proteins identified using SWATH MS analysis of MDA-MB-231 cell lysate and secretome following adrenoceptor agonism/antagonism.

	Lysate			Secretome		
Treatment (vs UT)	Protein	Swiss-Prot ID	Log2 Fold Change	Protein	Swiss-Prot ID	Log2 Fold Change
NE	No significantly changed proteins			STC1	P52823	1.385
				PPI1A	P62937	1.106
				CSF1	P09603	−1.025
				B2MG	P61769	−1.145
				LFNG	Q8NES3	−1.146
				PTX3	P26022	−1.151
				CUL5	Q93034	−1.160
				HUWE1	Q7Z6Z7	−1.172
				ANR28	O15084	−1.226
				UROK	P00749	−1.364
				ITIH2	P19823	−1.383
				CTGF	P29279	−1.443
				TSP1	P07996	−1.461
				KI13B	Q9NQT8	−2.489
ISO	G45IP	Q8TAE8	3.552	STC1	P52823	1.097
	ODBA	P12694	3.331	KI13B	Q9NQT8	−1.048
	BAF	O75531	−1.013	PPT1	P50897	−1.062
	DBNL	Q9UJU6	−1.025	CAD11	P55287	−1.088
	S10A4	P26447	−1.037	CSF1	P09603	−1.109
	THIO	P10599	−1.074	HUWE1	Q7Z6Z7	−1.164
	NEDD8	Q15843	−1.117	TSP1	P07996	−1.219
	ANXA2	P07355	−1.129	CTGF	P29279	−1.446
	CYTB	P04080	−1.171	UROK	P00749	−1.564
	PEBP1	P30086	−1.240	CATD	P07339	−2.309
	PRDX3	P30048	−1.273	ANR28	O15084	−2.313
	LEG1	P09382	−1.329			
	GLRX3	O76003	−1.526			
	ANXA5	P08758	−2.254			
	TGM2	P21980	−4.288			
ICI 118,551	GLSK	O94925	1.757	HUWE1	Q7Z6Z7	−1.053
	2A5E	Q16537	1.179	ANR28	O15084	−1.581
	LEG1	P09382	−1.017			
	PRDX3	P30048	−1.027			
	ANX11	P50995	−1.097			
	OST48	P39656	−1.141			
	ANXA2	P07355	−1.491			
	CALX	P27824	−1.565			
	S10AA	P60903	−1.615			
	ANXA4	P09525	−1.670			
	PEBP1	P30086	−1.788			
	GALT2	Q10471	−1.927			
	DNJA2	O60884	−2.297			
	CP1B1	Q16678	−3.311			
ICI 118,551 + NE	ODBA	P12694	3.341	RLA2	P62805	1.019
	S10AA	P60903	−1.077	CUL5	Q93034	−1.049
	ANX11	P50995	−1.223	H4	P62805	−1.071
	ANXA2	P07355	−1.460	TCRG1	O14776	−1.078
	ANXA4	P09525	−1.779	BGH3	Q15582	−1.079
	CP013	Q96S19	−2.699	PPT1	P50897	−1.175
				CYTS	P01037	−1.180
				TSP1	P07996	−1.230
				CYTN	P01037	−1.298
				SRPX	P78539	−1.345
				HUWE1	Q7Z6Z7	−1.454
				ITIH2	P19823	−1.509
				FINC	P02751	−1.510
				ANR28	O15084	−1.930
				KI13B	Q9NQT8	−2.080

Log2 fold change represents the increase/decrease in protein expression relative to untreated samples (UT). Proteins with a Log2 fold change of <1.0 or >1.0 and OneOmics confidence of ≥60% are shown. Proteins with single peptides are excluded. Data is from six biological replicates. NE—norepinephrine, ISO—isoproterenol, UT—untreated.

**Table 3 biology-09-00039-t003:** Clinical details of cases histologically assessed for LYPD3 protein expression was compared in malignant (primary and metastatic) breast adenocarcinoma tissue.

	LYPD3 Score		
	Negative	Positive	Chi Square (*p*-Value)
Age			
<40	55 (91.7%)	5 (8.3%)	2.885 (0.236)
40–59	179 (84.8%)	32 (15.2%)	
>60	43 (91.5%)	4 (8.5%)	
Tumour Grade			
1	39 (14.1%)	2 (4.9%)	10.034 (0.074)
1–2	4 (1.4%)	1 (2.4%)	
2	105 (37.9%)	19 (46.3%)	
2–3	0 (0)	1 (2.4%)	
3	64 (23.1%)	9 (22%)	
Tumour stage			
I	18 (6.5%)	2 (4.9%)	15.712 (**0.028**)
IIA	102 (36.8%)	8 (19.5%)	
IIB	32 (11.6%)	4 (9.8%)	
IIIA	5 (1.8%)	3 (7.3%)	
IIIB	19 (6.9%)	3 (7.3%)	
IV	5 (1.8%)	1 (2.4%)	
Oestrogen receptor status			
Negative	137 (69.2%)	16 (76.2%)	0.442 (0.506)
Positive	61 (30.8%)	5 (23.8%)	
HER2 status			
Negative	162 (81.4%)	14 (66.7%)	2.580 (0.108)
Positive	37 (18.6%)	7 (33.3%)	
Tissue pathology status			
Malignant primary breast tumour	219 (86.9%)	33 (13.1%)	3.535 (0.171)
Metastasis	39 (83%)	8 (17%)	
Adjacent normal breast tissue	19 (100%)	0 (0%)	

Significant values are shown in **bold** and missing data censored.

## Data Availability

The mass spectrometry proteomics data that support the findings of this study have been deposited to the ProteomeXchange Consortium [71] via the PRIDE partner repository [72] with the dataset identifier PXD009488. All other data generated or analysed during this study are included in this published article (and its Appendix A).

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
