# Peer review of "β2-Adrenergic Signalling Promotes Cell Migration by Upregulating Expression of the Metastasis-Associated Molecule LYPD3"

_biology, 2020, doi:10.3390/biology9020039_

Round 1

Reviewer 1 Report

The manuscript by Gruet et al., explores b2-adrenoreceptor-mediated cell behaviors and expression of LYPD3 in cancer. Previous studies showed inconsistencies in how beta-adrenergic receptor affects tumor proliferation. Using four different breast cancer cell lines, the authors showed that norepinephrine treatment enhanced metastasis-related cell behaviors including cell invasion. Subsequent proteomic work identified LYPD3 to be upregulated in one of breast cancer cell lines upon norepinephrine-treatment and increased LYPD3 expression was verified in several cancer types. Overall, the authors did well to study the role of b2-adrenoreceptor using a combination of in vitro experiments, proteomic studies and analyses of in vivo data. However, there are several major concerns and minor issues with the manuscript that the authors must address before this manuscript can be considered for publication:

Major concerns:

In the abstract, stating that “LYPD3 mRNA expression appeared to correlate with… EMT” is inappropriate as only four cell lines were tested and none of molecular markers of EMT was examined. Therefore, while such statement can be left in the discussion, it should be removed from the abstract. Fig 1D. The authors need norepinephrine alone treatment condition for this experiment. This is important as norepinephrine treatment is the condition which induces significant effects in subsequent figures. Also on Fig 1D. Since cells were assessed only at one-time point (72 h, according to methods), it would not be appropriate to use the term cell proliferation for this data. Thus, more suitable label for the y-axis would be cell survival, not growth. Also, this data is likely better suited as a supplementary figure for Fig 2. 2.1: It wasn’t clear from the text if the authors were testing cell-cell or cell-substrate adhesion. It’s only in the figure legend, that the authors specify that they are testing cell-substrate adhesion. The logic for assessing cell-substrate adhesion and using three different substrates need to be better introduced. Fig 2 is of very low resolution and labels are extremely difficult to make out. Not clear what the authors mean in line 111-113 as the presence of norepinephrine had virtually same effects on all cell lines. 2.2: Since BT-549 and MCF-7 cells do not show any statistical significance between different conditions, it would be good to show results from the scratch migration assays where the authors claim “ICI-118,551 completely abrogated the enhanced migration… induced by norepinephrine treatment.” Why MDA-MB-468 and MDA-MB-231, but not BT-549 and MCF-7, were chosen for mass spec is not explained (cells containing higher levels of ADRb2 mRNA and/or displaying most response to isophroterenol?). For Fig 3E, the authors should list proteins falling under each biological process. This can be shown as a supplementary table. Although the authors showed LYPD3 gene expression in Fig 3F, protein levels of LYPD3 among different cell lines should be shown, as mRNA and protein levels may not necessarily correlate (Fig 1A and B). Although MCF-7 has the highest expression of LYPD3, it is still surprising that the authors chose to examine cell migration using this cell line (Fig 3G) given the fact that according to Fig 2B, NE and ICI-118,551 had virtually no effect on MCF-7 cell migration. The authors should comment on the discrepancy between Fig 2B and 3G. Lines 329-330: Tumor aggression was not tested in this study. If the authors are referring to in vitro results, the authors can perhaps state instead “cell behaviors indicative of tumor aggression.” It is not clear what accounts for NE-stimulated effects in MDA-MB-231 in Fig 2. While the authors did mention EMT as a potential inhibitor of LYPD3 expression, the authors should describe how MDA-MB-231 may exhibit NE-stimulated behaviors. In light of this, using DAVID to analyze proteins differentially expressed in MDA-MB-231 and including this data in this manuscript would be informative.

Minor issues:

Fig 1A. The x-axis is very difficult to read with the font size being too small. The authors should consider making 1A wider while moving 1B to the second row with 1C. Fig 1C. The y-axis should indicate what is being examined (cAMP). Also this figure looks a bit odd with horizontal bars. The authors should consider showing the cAMP level on the y-axis and cell lines on the x-axis with control coming before isoproterenol for each cell line. Fig 1D. The font size for axes labels is on the small side. Line 438, there’s no reference following “as described previously.” Fig 2A. Representative images for this figure would be a nice addition to the supplementary data. Line 129: Reference should be provided for “what others in the literature have observed.” While Fig 2B shows migration as relative fluorescence intensity, Fig 3G shows migratory index. The authors should keep labels consistent if a same method was used. Fig 3A needs a scale bar. Fig 3D & F. Define HKG. Fig 3G. In the figure legend, “MCF-7” should be labelled as “MCF-7 vehicle” so that the figure and text are consistent. It seems that NE still induced migration in shLYPD3 cells. Therefore, statistical significance should be tested between +/- NE for shLYPD3 cells. The authors should also show protein levels of LYPD3 in MCF-7 vehicle, shControl, and shLYPD3 cells +/- NE.

Reviewer 2 Report

The authors demonstrate that β2-adrenergic signaling promotes cell migration by upregulating LYPD3 especially in the epithelial-derived cancer cells, but not in mesenchymal cell lines. They suggest that LYPD3 can be a promising therapeutic target in the treatment of cancers. The experiments are well-designed and the diverse approach to verify the results is impressive. However, several parts of the manuscript should be modified and more supporting data are needed to prove their hypothesis. I hope this article to be posted after major revision.

1. Please check a spelling of ‘signalling’.

2. In Figure 3C, why is the expression of LYPD3 increased compared to the untreated cells by ICI118,551 treatment?

3. In Figure 2B, why did NE and ICI-118,551 had no influence on the cell migration of BT-549 and MCF-7? In the same context, I wonder why ICI-118,551 didn’t inhibit the invasive activity of BT-549 cells in Figure 2C. There should be some explanation about these results.

4. In Figure 3G, the migration of MCF-7 cells was significantly increased by NE treatment. But In Figure 2B, the migration of MCF-7 cells was not altered by NE treatment. Which one is right?

5. In Figure 3, please add the full name of HKG in the figure legend.

6. In Figure 3G, please add significance stars in a graph between

shControl (–) vs. shControl (+) shControl (+) vs. shLYPD3 (+) shLYPD3 (–) vs. shLYPD3 (+).

     to clearly demonstrate that LYPD3 regulated by ABRβ signaling pathways is important for cancer cell migration.

7. Please conduct the same experiments of Figure 3G using MDA-MB-468 or MDA-MB-453 cells to confirm the migration-promoting function of LYPD3

8. In line 31, I don’t know the meaning of “in the context of cell morphology”. Is there any result showing that LYPD3 induced morphological changes of cancer cells from epithelial phenotype to mesenchymal phenotype? To say like this, you should present a microscopic data or at least show the expression of EMT markers by western blot.

Round 2

Reviewer 1 Report

The authors have adequately addressed most of concerns raised. However, some changes were incorporated rather awkwardly into the manuscript (for ex., section 2.2.2). Also, there are few additional suggestions in response to the authors' comments. Please refer to the attached document. New responses are highlighted in yellow.

Author Response

Thank you for your additional comments. I hope that our rebuttal below to the comments highlighted in the pdf are satisfactory.

The statement from the lines 111-113 (which are now lines 120-121) fell under the “Cell adhesion” section, which relates to Fig 2A. Therefore, if the statement was made in reference to the migration, the authors should move lines 120-121 to the appropriate section. In regards to this, it should be noted that the authors already made a reference to reduced migration of NE-treated MDA-MB-231 cells in lines 128-130. This was an error on our part. The statement was in reference to the migration and as you have correctly pointed out we have already made reference to the migration data and therefore this comment does not answer your original point. With regard to your original point, referring to lines which are now 120-121, the following sentence has been deleted as we feel that it does not make sense in this context “More importantly, that MDA-MB-231 cells depend on the presence of a substrate coated onto the tissue culture plastic in order to respond to norepinephrine stimulation (data not shown)”.

The authors provide an adequate explanation for the difference between Figs 2B and 3G. However, such comment did not appear in the revised manuscript. It is recommended that the authors point out the difference in the manuscript and at least mention the concentration difference as the likely culprit as the disconnect between Figs 2B and 3G will be rather obvious to ordinary readers. When left ignored, it can damage credibility of other aspects of this nice work. This is an oversight on our part as when responding to the reviewer’s comments we did not make it clear that the following was added into the manuscript at lines 227-231. “The increase in migration observed in MCF-7 vehicle cells is more than originally observed (Figure 2B) and this could be due to 2 factors; a lower concentration of norepinephrine was used to stimulate the cells (100 nM vs 10 µM) and the cells used in the knockdown studies were of a slightly lower passage number (passage 5 vs passage 8).”

Western blot lanes are not labelled. We have now labelled the western blot lanes in Supplementary Figure 5.

Reviewer 2 Report

I think the manuscript has been improved and now warrants publication in Biology.

Author Response

Thank you for accepting our previous revisions. You gave no further revisions to consider.